# A Systematic Review of Water, Sanitation, and Hygiene for Urban Poor in Low- and Middle-Income Countries during the COVID-19 Pandemic through a Gendered Lens

**DOI:** 10.3390/ijerph191911845

**Published:** 2022-09-20

**Authors:** Krushna Chandra Sahoo, Shubhankar Dubey, Girish Chandra Dash, Rakesh Kumar Sahoo, Mili Roopchand Sahay, Sapna Negi, Pranab Mahapatra, Debdutta Bhattacharya, Banamber Sahoo, Subhada Prasad Pani, Mariam Otmani del Barrio, Sanghamitra Pati

**Affiliations:** 1Health Technology Assessment in India (HTAIn), ICMR-Regional Medical Research Centre, Bhubaneswar 751023, India; 2Department of Psychiatry, Kalinga Institute of Medical Sciences, Bhubaneswar 751024, India; 3Department of Research and Development, Sri Balaji Vidyapeeth Deemed to be University, Puducherry 605007, India; 4UNICEF/UNDP/World Bank/WHO Special Programme for Research and Training in Tropical Diseases (TDR), World Health Organization, 1211 Geneva, Switzerland

**Keywords:** pandemic, COVID-19, WASH, urban poor, slum, gender, sanitation, water, LMICs

## Abstract

Inadequate water, sanitation, and hygiene (WASH) among urban poor women is a major urban policy concern in low- and middle-income countries (LMICs). There was a paucity of systematic information on WASH among the urban poor during the pandemic. We reviewed the opportunities and challenges faced by the urban poor in LMICs during the COVID-19 pandemic. We used the PRISMA guidelines to conduct a comprehensive search of 11 databases, including MEDLINE, Embase, Web of Science, and CINAHL, between November 2019 and August 2021. We used thematic analysis to synthesize the qualitative data and meta-analyses to estimate the pooled prevalence. We screened 5008 records, conducted a full-text review of 153 studies, and included 38 studies. The pooled prevalence of shared water points was 0.71 (95% CI 0.37–0.97), non-adherence to hygiene practices was 0.15 (95% CI 0.08–0.24), non-adherence to face masks was 0.27 (95% CI 0.0–0.81), and access to shared community toilets was 0.59 (95% CI 0.11–1.00). Insufficient facilities caused crowding and long waiting times at shared facilities, making physical distancing challenging. Women reported difficulty in maintaining privacy for sanitation, as men were present due to the stay-at-home rule. Due to unaffordability, women reported using cloth instead of sanitary pads and scarves instead of masks.

## 1. Introduction

Target 6.2 of the Sustainable Development Goals (SDGs) calls for universal access to adequate and equitable sanitation. In 2010, the United Nations General Assembly recognized access to safe and clean drinking water and sanitation as a human right. It urged international cooperation to assist countries in establishing safe, clean, accessible, and affordable drinking water and sanitation systems [1]. Around two billion people still lack adequate sanitation, such as toilets or latrines. Of these, 673 million continue to use open defecation (OD). Inadequate sanitation contributes to infection transmission and impedes social and economic growth [2]. Additionally, approximately 827 thousand people die each year in low- and middle-income countries (LMICs) due to insufficient water, sanitation, and hygiene (WASH) [2,3]. As a result, from an environmental health viewpoint, improved WASH is a high-priority public health intervention in LMICs.

The world has become increasingly urbanized, with most of this growth occurring in LMICs, such as those in sub-Saharan Africa [4]. Inadequate WASH performance is a primary urban policy concern in cities of LMICs, such as Kenya, India, and Indonesia, particularly among the urban poor, including the homeless, refugees, and informal settlements. Informal settlements are defined by overcrowding, inadequate infrastructure, and a lack of social amenities [5,6,7,8,9]. According to a study in India, sanitation behaviors go beyond only urinating and defecating to include fetching water, cleaning, taking baths, managing menstrual cycles, and changing clothes. Women engaged in these activities face environmental, social, and sexual stressors that vary depending on the woman’s phase of life, her housing situation, and her access to toilet facilities [8]. Women and girls were primarily responsible for looking after the WASH needs of the households. In sub-Saharan Africa, women and girls spend most of their day hours fetching water to meet their water requirements [10]. Despite evidence of an elevated risk of ill health consequences compared with individual household latrines, a sizable and growing proportion of the global population relies on shared sanitation facilities. According to a systematic review, individuals that are dependent on community toilets have an increased risk of diarrhea infection and experience more sanitation-related stress [8,11,12]. Safe WASH services, including a primary water supply, sanitation, and hygiene, are crucial for the urban poor.

WASH services are crucial for preventing and protecting human health during pandemics, such as the current COVID-19 pandemic [3,13]. Additionally, it is important to ensure the viability of crucial supply chains—soap, disinfectant, and point-of-use water treatment—and enforce import/export limits on critical home equipment [14]. Communication and preparation for behavior change and promotion of handwashing and safe drinking water practices were also essential in preventing and spreading the COVID-19 pandemic [13,15]. Women and girls have unique sanitary requirements compared to men [16]. Women spend substantially more time on unmet WASH needs than men, which may cause severe challenges during the COVID-19 pandemic, particularly in patriarchal societies [17]. However, there is a lack of systematic data on WASH-related evidence among the urban poor in LMICs during the COVID-19 pandemic. As a result, this study explored the WASH-related opportunities and constraints experienced by the urban poor in LMICs during the COVID-19 pandemic from a gendered perspective, focusing on access to water and toilet facilities, WASH responsibilities, and hygiene practices, including hand hygiene and mask use, during the COVID-19 pandemic.

## 2. Methods

The systematic review and gap analysis were carried out following the PRISMA guidelines [18]. The protocol was registered in PROSPERO (CRD42021292456). In the context of the COVID-19 pandemic, we included studies on WASH among slum dwellers and homeless populations in urban regions of LMICs.

### 2.1. Databases, Search Strategies, and Selection Process

We conducted a comprehensive search of the following eleven databases for relevant articles published between November 2019 and August 2021: PubMed (MEDLINE), Embase, Web of Science, WHO Global Index Medicus, Epistemonikos, ProQuest, CINAHL, Cochrane Library, medRxiv and bioRxiv, 3ie, and Google Scholar.

“COVID-19” AND “Urban poor” AND “LMICs” was used as a search strategy, together with all associated keywords and MeSH terms. After identifying all records using the aforementioned searches, we screened all primary studies on WASH. All countries on the World Bank’s list of LMICs [19] were taken into account. Only primary studies—qualitative, quantitative, and mixed-methods—were considered. We did not limit our research to a specific language. We did not include reviews, opinions, perspectives, or editorials.

All identified citations were entered into Endnote X8 (Clarivate, PA, USA), and duplicate records were removed. Following the removal of duplicates, the remaining articles were imported into Rayyan, which is a web-based application for title and abstract screening. Using Rayyan, reviewers (S.D., R.K.S., M.R.S., S.N.) screened the studies based on their titles and abstracts. Furthermore, the full texts of the potential studies were retrieved and manually screened to ensure their eligibility for selection. The authors conducted the entire screening process independently and in duplicate. Any disagreements that were raised during the study selection process were resolved through discussion and mutual agreement with the reviewers (K.C.S., G.C.D.).

### 2.2. Quality Assessment, Data Extraction, and Synthesis

Authors (K.C.S. and M.R.S.) assessed the quality of the selected studies, which were assessed using the Mixed Methods Appraisal Tool [20] (Appendix A). Disagreements were sorted through discussion and mutual consensus among authors (K.C.S., G.C.D.).

Reviewers (S.D. and K.C.S.) extracted quantitative data in a Microsoft Excel sheet comprising the study characteristics: author, year, country, types of the urban poor, total population, study type, data collection methods, and major WASH domain of each study (Table 1). We resolved disagreements at any stage through discussions between authors.

Furthermore, an author (K.C.S.) reviewed all the studies, open-coded the information, and prepared a codebook; a thematic framework emerged from the data for selective coding in MAXQDA Analytics Pro 2020 (VERBI GmbH, Berlin). An author (S.D.) extracted all the information in MAXQDA using the selective coding approach. Finally, authors (K.C.S., S.D.) synthesized the results and prepared the results using thematic framework analysis. An author (G.C.D.) estimated the pooled prevalence (meta-analysis) from the available data using MetaXL Software Version 5.3 (EpiGear International Pty Ltd., Brisbane, Australia).

## 3. Results

We identified 6490 studies, including 1482 duplicates. Based on the titles and abstracts, 5008 studies were screened, resulting in 153 potential studies for full-text review. Out of these 153 studies, 38 met the eligibility criteria and were finally included in the review. The PRISMA flow diagram is provided to illustrate the entire study selection process in Figure 1.

### 3.1. Gender-Related Considerations Regarding Access to Water Facilities and Hygiene Practices during the COVID-19 Pandemic

Access to an adequate water supply was critical for maintaining hygiene practices during the COVID-19 pandemic. Most urban slum households in LMICs obtain their drinking water from public standpipes that operate on a limited schedule. Few slum dwellers have their own water supply (Table 2). Over half of the households indicated that they drank, bathed, and cleaned using communal water sources outside the home [25,51]. The pooled prevalence of shared water points as a water source among the urban poor was 0.71 (95% CI 0.37–0.97) (Figure 2).

According to a few studies, women were primarily responsible for household hygiene practices and water collection. In Dharavi, India, which is one of the largest slums across the globe, women took a more key role than men in household activities, such as fetching water and caring for children [39]. Women were typically responsible for water collection, which required frequent departures from their homes. As a result, many urban poor women struggled to collect water during the COVID-19 pandemic. Women typically purchase water in Indonesian slums from traveling vendors [39].

Many people are forced to go outside because they do not have running water in their homes, even though they understand that coming to a crowded place for water puts their lives at risk during the COVID-19 pandemic. When the water arrives late, there is a longer waiting period and a greater risk of exposure [47,50]. Inadequate water supply and long wait times in congested lanes increase the chances of SARS-CoV-2 transmission [21,22,31,35,39,45,51]. Water collection from shared and crowded sources was one of the primary sources of infection among urban poor women compared with men. Furthermore, due to the summer lockdown and increased demand for water due to the COVID-19 pandemic, there were water shortages, limiting the ability to wash hands [39,40,58].

Concerning hygiene behavior, although many of the slum-dwellers reported lacking access to sufficient water, they still carried out basic preventative procedures, such as washing hands more often and avoiding shaking hands [22,48,52,56]. The pooled prevalence of non-adherence to hygiene practices among the urban poor during the COVID-19 pandemic was found to be 0.15 (95% CI 0.08–0.24) (Figure 3, Table 2). During the COVID-19 pandemic, hygiene practices were limited by the lack of space, scarcity of water, and inability to purchase disinfectants [30,33,35,40,56]. Due to the lockdown, more than half of the urban poor reported not washing their hands with soap or even using hand sanitizer. Many urban poor shared that they used to buy water for 20 INR every day [40].


*During this pandemic, the majority of the community members are experiencing financial hardship. They barely have enough money to eat twice a day. How will they get soap to wash their hands?*


In Ethiopia, two-thirds of drivers who lived in urban slums routinely washed their hands with water without any soap. However, effective hand hygiene practices were substantially related to educational attainment and attitudes toward COVID-19 [36]. Similarly, a study conducted in Bangladesh discovered that the lower one’s income, the more agreement there was on risk reduction strategies due to treatment costs [54]. In Kenya, handwashing and the use of hand sanitizer were recognized as preventative measures. In contrast, the lack of a personal water supply (37%) and the high cost of hand sanitizer (53%) were barriers [25]. In Bangladesh, several urban poor shared that they could even use “chai” (ashes) as a replacement for the scarce soap/hand wash [41]. In Bangladesh, the transgender community practiced preventative measures, such as frequent hand, face, and foot washing and, most significantly, “no touching” and body fluid interchange [34].


*I cannot imagine how residents would practice physical distancing and hygiene in a crowding environment and having insufficient water.*


The pooled prevalence of non-adherence to wearing face masks as a measure of hygiene and prevention among the urban poor during the COVID-19 pandemic was found to be 0.27 (95% CI 0.0–0.81) (Figure 4, Table 2). In India, around 94% were aware of the importance of wearing a mask, social distancing, and the use of hand sanitizer; 94% avoided handshaking and hugging, but only 68% used alcohol-based hand sanitizer [55]. In Thailand, 96% of urban poor people wear face masks when they leave the house, and 92% use alcohol gel or wash their hands with soap whenever they touch something. They mentioned that masks and alcohol gel are freely available from government agencies and donors [46]. With the pandemic outbreak, many South Africans were forced to choose between purchasing protective masks and hand sanitizers and purchasing daily necessities, such as bread [29]. Because many urban poor cannot afford N95 masks, they use low-cost compensatory strategies, such as covering their faces with a simple scarf and washing them with water. A study found that approximately 37% of respondents reused their masks after cleansing with either water or soap water in India, and 9% admitted to occasionally sharing their masks with other family members [53]. Due to a scarcity of face masks, many women used their scarves/shawls as an alternative, while men wore masks. However, they often wash their scarves and cloth masks in boiling water with Dettol. They believed that because men frequently went outside to arrange food, they should wear masks to avoid infection [21,22,56]. Many women were taught to stitch masks for their community and earn money to support their families [27,32].

### 3.2. Considering Access to Basic Sanitation Facilities during the COVID-19 Pandemic with a Gendered Lens

Almost all studies found a daily lack of access to appropriate sanitation; however, the COVID-19 pandemic added to the difficulties. The pooled prevalence of access to shared community toilets was found to be 0.59 (95% CI 0.11–1.00) among the urban poor (Figure 4, Table 2). Households with private toilets were more likely to report an increased frequency of handwashing with soap following the lockdown. The absence of private toilets leads to the requirement to vacate the premises during a lockdown [30,49].

In Bangladesh, about 90% of slum toilets/baths/tube wells were shared by Rohingya community members residing in refugee camps [21]. Thus, distance, time, and location difficulties were deemed to be highly risky, as these increased the chances of contact. Many slums had no hand-washing facilities within five meters of toilets. They queued for up to half an hour, forcing the physical distancing and establishing a transmission hotspot [21,22,31]. The families lived in one- or two-room dwellings and shared bathrooms [30,41]. A study in Syria found four toilets for 270 tents. Those who share both toilet and bathing facilities (median = 2.9 m^2^/person, 95% CI 2.63–2.93) had the most crowded dwellings (median = 3.34 m^2^/person, 95% CI 2.79–3.34), while those who had sole use toilet and bathing facilities (median = 5.73 m^2^/person, 95% CI 4.18–6.5) had the least [51]. Due to the lack of supplies and inability to function, many individuals were forced to wait in a small area. During lockdowns, toilet usage doubled from pre-COVID-19 levels, it peaked during the morning and noon, when the wait time reached 25 min [21,22].

Due to space constraints, the number of people sharing the same space, and long lines at the communal toilets during the COVID-19 pandemic, it is challenging to practice physical distancing [39,41,43,47,49,51]. Although public toilets are free to use, they are poorly maintained and children avoid them [53]. The mobile toilets were filthy, and walking to the restroom took over half an hour, making it challenging for women and children [37]. Furthermore, most latrines were broken, and cleaning was infrequent. The interior water supply and handwashing facilities were non-existent, unclean, and hazardous [31,34,42,52,58].

The majority of residents also defecate in open areas adjacent to their communities [39]. While some men in the community walk large distances for open defecation (OD), women face greater obstacles since they are fearful of assault and do not choose to walk long distances for OD. Additionally, the stay-at-home restriction was depressing for both men and women who practice OD. Obtaining a proper location for OD was another apparent difficulty [26,28,37]. Around a third stated that they or a family member had to leave their house daily to defecate due to a crowd at the community toilet [49].

Women bear a disproportionate share of the burden of poor sanitation and were at increased risk of contracting SARS-CoV-2. The slum dwellers had equal access to latrines, and the majority of them used latrines that were safe for women and children at night [39,58]. Many community leaders reported that the camps’ latrines that were specific for men or women were in disrepair. Many women reported that because there were more men present during the COVID-19 pandemic, they could not maintain privacy and could not defecate on time, they felt physically unfit, and they drank less water, reducing the frequency with which they needed latrines [21]. Women who use public toilets face an increased risk of SARS-CoV-2 exposure due to the filthiness of the toilets and associated surfaces, as well as their physical proximity to other users [39]. Many women reported that they previously used pads but switched to clothes during the pandemic due to the cost of the pads [23,26,38].

### 3.3. Coping and Maladaptive Behaviors Associated with Poor Access to Water and Sanitation Facilities

Water for daily activities was difficult to obtain throughout the pandemic. The urban poor, who were reliant on supply water, were obliged to congregate or queue for extended periods to collect water. Adhering to COVID-19 standards and performing frequent hand washing became more difficult since many families experienced days without water due to the unpredictable and restricted duration of the water supply [21,22,44,50]. The slum residents sought the assistance of city officials in resolving the issue. They ordered water from tanks [50]. While water tankers can assist in the short term, long-term actions to boost water availability and resources will be required. Tanker water, on the other hand, is only economically viable for residents who can afford it [39,40,58].

Many urban poor women in Bangladesh lowered their water consumption to avoid walking long distances, crowding at water taps, and frequent visits to community latrines [21]. According to a study conducted in India, the lack of a community toilet forces people to engage in harmful behaviors, such as open defecation and withholding defecation, and conflict and violence often occur [24]. Children were forced to defecate in open drains in many camps due to a shortage of clean facilities. Another study conducted in India found that adequate water supply and quarantine facilities in refugee camps aided in the successful control of SARS-CoV-2 infection [27]. On the other hand, a South African study found that the primary concerns expressed by informal settlement inhabitants during the COVID pandemic were maintaining physical distance, self-isolation, and quarantine. Overcrowding of public toilets and a scarcity of basic amenities have deteriorated people’s lifestyles [29]. “Sometimes I have to skip showering to preserve water for cooking,” a woman in Mumbai, India, said. “Yet you want us to wash our hands all the time?” Certain individuals dug their toilets in an extremely inefficient manner, resulting in a constant overflow of sewage waste [31].

### 3.4. Support System and Community Engagement to Improve Access to Water and Sanitation

The educated and skilled community took the initiative to restrict gatherings at the community bathrooms [41]. Along with awareness programs, foot-operated devices for utilizing the sink and toilet were built. The public restrooms were disinfected and fumigated twice [42]. Hand-washing stations were erected, soap was provided, and clean water was stored, in addition to the WASH sector’s distribution of hygiene promotion messaging to ensure effective WASH practice [57].

Slum leaders attempted to secure water availability by communicating with superiors and occasionally contacting private water tankers [50]. Regular delivery of piped water to communities allowed for the practice of hand hygiene using soap [53]. In numerous instances, non-governmental organizations (NGOs) and local governments aided WASH practices by supplying masks and sanitizers [38]. NGOs and aid organizations frequently placed communal tube wells, taps, and toilets in vacant areas along roadways, frequently without regard for the distance and time required for everyone, and in violation of COVID-19 restrictions [21,31,41]. Each dwelling in the refugee camps received water via house taps or community taps, which enabled them to effectively implement WASH standards [27]. However, many reported purchasing water for drinking and hand hygiene routines or having it facilitated by the government’s water supply via water tankers [27,39]. NGOs’ efforts to promote and upgrade WASH were futile in the context of the COVID-19 pandemic. The provision of free hygiene kits containing soaps, sanitizers, tissue papers, and masks to contain infection ceased during the pandemic [31,47,53,54,58].

## 4. Discussion

This review highlighted the challenges related to WASH among the urban poor men and women in LMICs during the COVID-19 pandemic. This review revealed that the majority of the urban poor population used shared water points and toilets in their daily lives, posing unique challenges for men and women during the COVID-19 pandemic. This study showed that the pooled prevalence of shared water points was found to be 0.71 (95% CI 0.37–0.97), non-adherence to hygiene practices was found to be 0.15 (95% CI 0.08–0.24), non-adherence to face masks as a measure of hygiene and prevention was found to be 0.27 (95% CI 0.0–0.81), and access to shared community toilets was found to be 0.59 (95% CI 0.11–1.00) among the urban poor. Inadequate water and sanitation facilities, which produce crowds and long wait times in congested lanes at shared facilities, make physical distancing difficult to practice during the COVID-19 pandemic. Due to the lockdown rule, all slum dwellers, particularly men, were forced to stay at home and depend on shared toilets and bathing facilities. Furthermore, because the OD point was located a long distance away, the community members were unable to reach it due to the stay-at-home regulation. Women frequently noted that because there were more men present during the COVID-19 pandemic, they were unable to maintain privacy, and thus, withheld defecation and urine. As a result, they reduced their water intake to avoid using the shared community toilet. The women indicated that they used cloth instead of sanitary pads during the COVID-19 pandemic due to the high cost and scarcity of sanitary pads. Due to a paucity of face masks, many urban poor women used scarves/shawls instead, while men wore masks. Many women were trained in the stitching of masks for their communities.

Community-led committees should be engaged, trained, and equipped to advocate for the importance of isolation, an alternate method of handwashing, and alternative physical distance. Hand sanitizers containing alcohol can be used in handwashing [59,60]. COVID-19 mandates further water use for handwashing and improved hygiene, compounding the financial hardship households have regarding paying for water [39,40]. In a pandemic emergency, internal water supplies and handwashing facilities can help to prevent infection [58]. The pre-lockdown levels of water supply and sanitation appear to have remained constant in the majority of localities. It is critical to highlight that many urban poor lacked adequate water and sanitary services before the lockdown. This insufficiency remained for many residents during the lockdown [50]. Due to the high population density and overcrowding in slums, the risk of illness spreading is more significant than in a less crowded community. Several studies demonstrated that slum inhabitants are forced to live in filthy conditions due to poor sanitation. Compared to men, women face the burden of lack of access to water and sanitation and the risk of exposure to SARS-CoV-2 as a result of WASH-related activities [39]. Due to a lack of access to water and sanitary facilities, this recommendation is difficult to follow in slums.

The number of users per latrine is critical since it is necessary to maintain the highest level of physical distancing. This is worsened because a sizable fraction of slum households are forced to travel outside their communities to obtain necessities such as water and sanitation. Due to the lack of water and sanitary facilities in slums, it is difficult to adhere to this regulation [40]. Handwashing and hygiene materials include the provision of fixed and portable handwashing facilities, the procurement of soap and alcohol-based hand rubs, the provision of handwashing water sources, and point-of-use water treatment. WASH activities are required to ensure an effective COVID-19 pandemic response. Education-wise, slum dwellers hold divergent views on several COVID-19 risk mitigation measures, including limiting sunlight exposure; increasing physical activity; constantly washing hands, face, and feet; wearing a mask; washing hands with soap water or hand sanitizer; and gargling with hot water [54].

The COVID-19 pandemic has had a substantial worldwide impact on the livelihoods, health, and general well-being of the urban poor in LMICs, where WASH insecurity is common and inextricably linked to vulnerabilities [10]. COVID-19 disproportionately impacts the world’s most vulnerable individuals, most of whom live in informal settlements and rural communities [17,21,22,31,35,39,45,51]. In most LMICs, the gender distribution of labor places women in charge of family well-being, health, and cleanliness. The burden associated with this obligation is exponentially more significant in households and communities without access to clean, safe, inexpensive, and accessible water and toilets [8,11,12,16]. The women faced sanitation challenges daily due to the community toilet being located a considerable distance away. However, during the COVID-19 pandemic, with all community members staying at home due to the lockdown, they were forced to depend on the public/community toilet, resulting in overcrowding, lengthy wait times, and WASH-associated conflict and violence at community facilities [8,21,22,33,35,40]. It is practically impossible to maintain social/physical distance in congested slums where people live nearby and share facilities [57]. Additionally, many respondents reported using public bathrooms, kitchen facilities, and bathing areas, which is a constant aspect of slum life. The majority of respondents were well aware of the increased risk of infection associated with the usage of communal facilities but indicated that they did not have a choice [41,59,61].

There are several ways in which the WASH sector has attempted to integrate gender-related considerations and women’s rights into one of the most crucial development and humanitarian sectors. Inadequate access to safe drinking water and sanitation exacerbates the impact of COVID-19 on women and girls living in urban poverty in LMICs [10,60]. The COVID-19 pandemic has a particularly severe effect on women and girls since they shoulder a disproportionate share of the burden of water collection, sanitation, hygiene, and family welfare, all of which are rooted in long-standing societal norms. WASH insecurity is a term that relates to physical and relational disparities in access to WASH. Women and girls have unique sanitation needs compared to men [10]. Women who fail to achieve this are frequently punished, sometimes violently. Along with their own washing needs, women bear responsibility for the WASH needs of others. Over the last four decades, there has been a growing recognition that social and political marginalization and complex disparities, including gender inequality and health inequity, are the primary drivers of apparent economic and technological disadvantage [16].

## 5. Conclusions

This review depicted the real picture of the urban poor and their vulnerability due to their risky living conditions, lack of state identity, and ambiguity about their living standards. In LMICs, despite the additional vulnerability posed by the pandemic, they fought the situation diligently with the limited WASH resources available. The urban poor need a support system to enhance their resilience capacity. All stakeholders, public and private, are therefore required to adopt a collaborative approach to manage such an unprecedented crisis. This review also revealed a requirement for sustained activation of gender-responsive strategies regarding WASH.

Women can legitimately be viewed as a preliminary step for reform efforts to comprehend local established practices because they help lower the health concerns for their households through their contextual knowledge and unpaid labor. Initiatives should strike a delicate balance between recognizing and enhancing women’s understanding, as well as involvement in influencing hygiene behaviors, while minimizing the gender discriminatory burden associated with carrying out these activities. Furthermore, collaboration with community leaders to develop peer education programs is necessary to improve gender-related consideration in WASH behavior and practices, particularly during emergencies in slum settings.

Women’s active participation in community WASH, on the other hand, is critical for achieving SDG 6.2, which calls for universal access to adequate and equitable sanitation. However, this review indicated limited evidence on WASH among the urban poor during an emergency from a gendered perspective. There is an urgent need for researchers to conduct more rigorous research on WASH among the urban poor cohort while taking into account the gender dynamics, not just confined to LMICs, but globally. 

## Figures and Tables

**Figure 1 ijerph-19-11845-f001:**
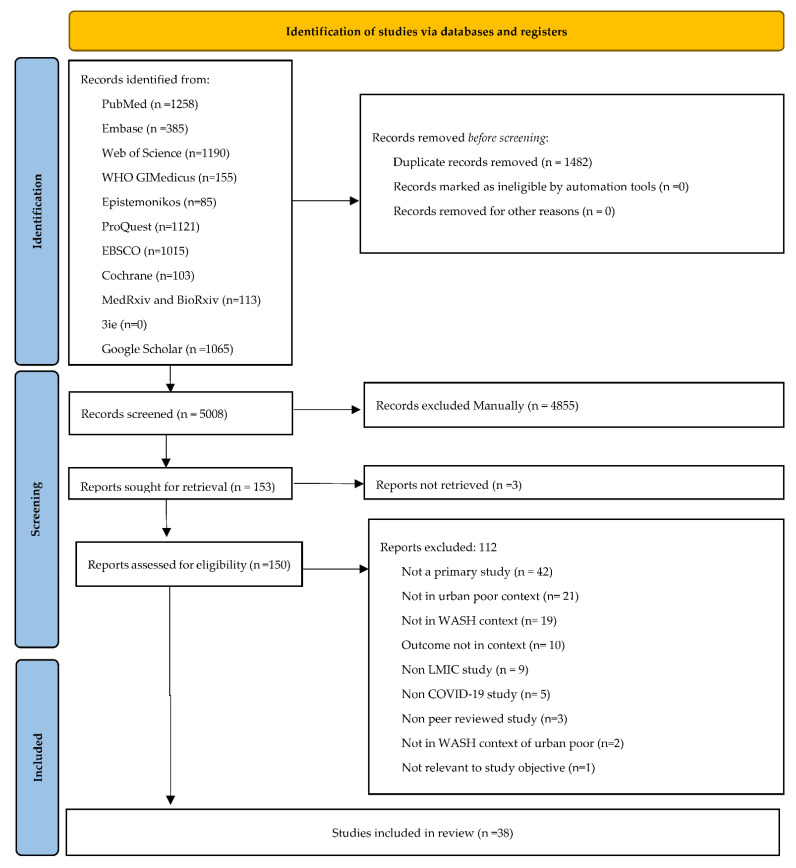
PRISMA flow diagram.

**Figure 2 ijerph-19-11845-f002:**
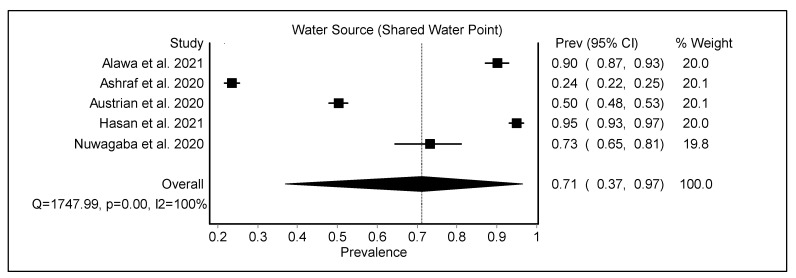
The pooled prevalence of shared water points for water sources among the urban poor [25,48,49,51,56].

**Figure 3 ijerph-19-11845-f003:**
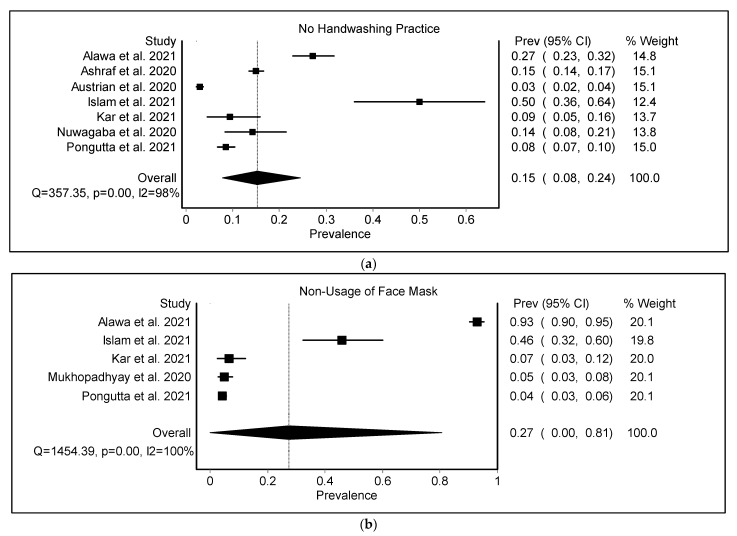
(**a**) The pooled prevalence of non-adherence to hygiene practices among the urban poor during the COVID-19 pandemic. (**b**) Pooled prevalence of non-adherence to face masks as a measure of hygiene and prevention among urban poor during the COVID-19 pandemic [25,46,48,49,52,53,55,56].

**Figure 4 ijerph-19-11845-f004:**
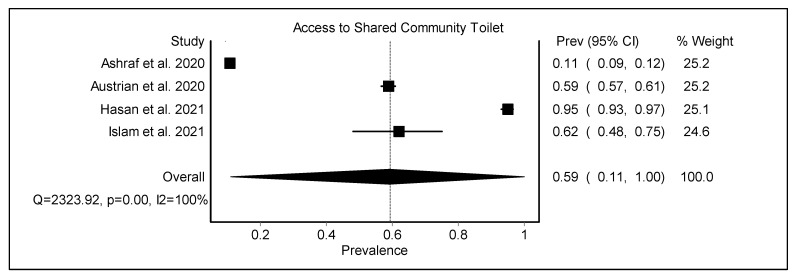
The pooled prevalence of access to shared community toilets among urban poor [25,49,51,52].

**Table 1 ijerph-19-11845-t001:** Study characteristics.

Author, Year	Country	Urban Poor	*N*	Study Type	Data Collection	Major WASH Domains
Akter et al., 2021 [21]	Bangladesh	Refugee	66	Qualitative	In-depth interviews	Water, sanitation, and Hygiene
Akter et al., 2021 [22]	Bangladesh	Slum dwellers	42	Qualitative	In-depth interviews	Water, sanitation, and Hygiene
Amdeselassie et al., 2020 [23]	Ethiopia	Slum dwellers	16	Qualitative	In-depth interviews	Water and hygiene
Arora et al., 2021 [24]	India	Women migrant worker	5	Qualitative	In-depth interviews	Sanitation
Austrian et al., 2020 [25]	Kenya	Informal settlements	2009	Qualitative	Survey: mobile phones	Water and hygiene
Azeez et al., 2020 [26]	India	Migrant women	19	Qualitative	In-depth interviews	Menstrual hygiene
Bercegol et al., 2020 [27]	India	Slum	-	Qualitative	Telephonic in-depth interviews	Water, sanitation, and hygiene
Bhattacharya et al., 2021 [28]	Bangladesh	Urban poor	-	Qualitative	In-depth interviews	Water
Cloete et al., 2020 [29]	South Africa	Sex workers and homeless	60	Qualitative	Informant interview, focus group discussion	Water, sanitation, and hygiene
Collantes et al., 2021 [30]	Philippines	Informal communities	-	Qualitative	Case study	Water and hygiene
Douedari et al., 2020 [31]	Syria	Camp residents	20	Qualitative	In-depth interviews	Water and hygiene
Ebekozien et al., 2021 [32]	Nigeria	Informal settlement	40	Qualitative	In-depth interviews	Water and hygiene
Iwuoha et al., 2020 [33]	Nigeria	Suburban slums	49	Qualitative	In-depth interviews and observation	Water and hygiene
Jalil et al., 2021 [34]	Bangladesh	Hijra community	22	Qualitative	Telephonic in-depth interviews	Sanitation and hygiene
Munajed et al., 2020 [35]	Syria	Refugee	11	Qualitative	In-depth interviews	Hygiene
Natnael et al., 2021 [36]	Ethiopia	Taxi drivers	417	Qualitative	In-depth interviews and observation	Hygiene
Nyashanu et al., 2020 [37]	South Africa	Informal settlement	30	Qualitative	In-depth interviews	Sanitation
Oluoch-Aridi et al., 2020 [38]	Kenya	Informal settlement	71	Qualitative	Telephonic interviews	Hygiene
Parikh et al., 2020 [39]	India, Indonesia	Informal settlement	-	Qualitative	Focus group discussion and transect walks	Water and hygiene
Patel, 2020 [40]	India	Slums	-	Qualitative	Media reports analysis	Water and hygiene
Rashid et al., 2020 [41]	Bangladesh	Slums	51	Qualitative	Telephonic in-depth interviews (IDIs)	Water, sanitation, and hygiene
Sahu et al., 2020 [42]	India	Slum	-	Qualitative	Document analysis	Water and sanitation
Saldanha, 2021 [43]	India	Slum	1	Qualitative	Narrative	Water and sanitation
Napier-Raman et al., 2021 [44]	India	Slum	122	Mixed-Method	Rapid survey	Water
Peteet et al., 2020 [45]	India	slums	87	Mixed-Method	Semi-structured interviews	Hygiene
Pongutta et al., 2021 [46]	Thailand	Urban slums	900	Mixed-Method	Semi-structured interviews	Hygiene
Wasdani et al., 2020 [47]	India	Slum	6	Qualitative	Case study	Sanitation
Alawa et al., 2021 [48]	Somalia	Internally displaced people	401	Quantitative	Cross-sectional survey	Water, sanitation, and hygiene
Ashraf et al., 2020 [49]	India	Migrant workers	2657	Quantitative	Surveys: telephonic	Sanitation and hygiene
Auerbach et al., 2020 [50]	India	Slum	321	Quantitative	Telephonic survey	Water and sanitation
Hasan et al., 2021 [51]	Bangladesh	Urban slum	588	Quantitative	Survey	Water and sanitation
Islam et al., 2021 [52]	Bangladesh	Slum	1303	Quantitative	Online survey	Water, sanitation, and hygiene
Kar et al., 2021 [53]	India	Urban Slum	106	Quantitative	Semi-structured questionnaire	Hygiene
Mamun et al., 2020 [54]	Bangladesh	Slum	434	Quantitative	Survey	Hygiene
Mukhopadhyay et al., 2020 [55]	India	Slum	282	Quantitative	Telephonic survey	Hygiene
Nuwagaba et al., 2020 [56]	Uganda	Slum	359	Quantitative	Structured questionnaires	Water and hygiene
Shammi et al., 2020 [57]	Bangladesh	Refugees	-	Quantitative	Field survey	Water, sanitation, and hygiene
Shermin et al., 2021 [58]	Bangladesh	Slum	1134	Quantitative	Survey	Sanitation

**Table 2 ijerph-19-11845-t002:** Access to water, sanitation, and hand hygiene practices among the urban poor during the COVID-19 pandemic in LMICs.

Author, Year	Country	Population (Urban Poor)	Study Design	Total Urban Poor (N)	Access to Water During the COVID-19 Pandemic			Access to Sanitation during the COVID-19 Pandemic			Hygiene Practices Related to COVID-19		
					Supply of Water to a Common Shared Point *n* (%)	Supply of Water to Household *n* (%)	No Supply of Clean Water *n* (%)	Public/Community Toilet *n* (%)	Individual Household Toilet, *n* (%)	Open Defecation n (%)	Lack of Access to Soap n (%)	Lacked Use of Mask n (%)	No Handwashing Practices *n* (%)
Alawa et al., 2021 [48]	Somalia	Internally displaced	Cross-sectional	401	362 (90)	39 (10)	0	NR	NR	NR	256 (64)	373 (93)	109 (27)
Ashraf et al., 2020 [49]	India	Migrant workers	Cross-sectional	2044	481 (24)	1563 (76)		220 (11)	1235 (60)	589 (29)	NR	NR	307 (15)
Austrian et al., 2020 [25]	Kenya	Informal settlements	Cross-sectional	1811	911 (50)	900 (50)	0	1178 (59)	718 (36)	105 (5)	NR	NR	58 (3)
Hasan et al., 2021 [51]	Bangladesh	Slum	Secondary data	588	559 (95)	29 (5)	0	559 (95)	29 (5)	0	NR	NR	NR
Islam et al., 2021 [52]	Bangladesh	Slum	Online survey	50	NR	NR	NR	31 (62)	19 (38)	0	NR	23 (46)	25 (50)
Kar et al., 2021 [53]	India	Urban Slum	Cross-sectional	106	NR	NR	NR	NR	NR	NR	10 (9)	7 (7)	10 (9)
Mukhopadhyay et al., 2020 [55]	India	Slum	Cross-sectional	282	NR	NR	NR	NR	NR	NR	NR	14 (5)	NR
Nuwagaba et al., 2020 [56]	Uganda	Slum	Cross-sectional	112	82 (73)	0	30 (27)	NR	NR	NR	47 (42)	NR	16 (14)
Pongutta et al., 2021 [46]	Thailand	Slums	Cross-sectional	900	NR	NR	NR	NR	NR	NR	NR	38 (4)	76 (8)

## Data Availability

The datasets generated during and/or analyzed during the current study are available from the corresponding author on reasonable request.

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
