# Peer review of "A Systematic Review of Water, Sanitation, and Hygiene for Urban Poor in Low- and Middle-Income Countries during the COVID-19 Pandemic through a Gendered Lens"

_ijerph, 2022, doi:10.3390/ijerph191911845_

Round 1

Reviewer 1 Report

This type of research has its contribution and is encouraged.

The study was conducted based on the PRISMA procedure, and data were collected from 11 databases.

Overall, this article is well-structured and provides a lot of information. However, be sure to strengthen the layout of the article, and also enhance the resolution of graphs and tables.

This is your own article. It is recommended to use your own style of PRISMA Flow Diagram. Do not just follow.

Author Response

Reviewer 1

RC: This type of research has its contribution and is encouraged. The study was conducted based on the PRISMA procedure, and data were collected from 11 databases.

AR:  Thank you.

RC: Overall, this article is well-structured and provides a lot of information. However, be sure to strengthen the layout of the article, and also enhance the resolution of graphs and tables.

AR:  Thank you. We have rechecked the layout of the article, and the resolution of graphs and tables.

RC: This is your own article. It is recommended to use your own style of PRISMA Flow Diagram. Do not just follow.

AR: Thank You.

Reviewer 2 Report

The manuscript makes an important literature contribution regarding the problems of water and sanitation in low income or poor countries. This is because inadequate water, sanitation, and hygiene (WASH) among urban poor women is a major urban policy concern in low- and middle-income countries (LMICs).  Moreover, a gendered-perspective is provided in the manuscript  which adds to the theoretical lens in which the research problem is examined. While the research reported is original and written relatively clear there are a few conceptual problems in a few areas.  A list of comments and suggestions is given for improvements and they are summarised as follows: 

Comments and Suggestions for Improvement:

(1) The original title is stated as 'A systematic review and gap analysis of opportunities and  challenges in Water, Sanitation, and Hygiene (WASH) for urban  poor in low- and middle-income countries during the COVID- 4  pandemic through a gender lens'. Given the word count (i.e about 33 words) in this article, this  title is too long. I suggest that you cut out the abbreviation (WASH)  to at least try to shorten it and remove other superflous words.  

(2) Your introduction reads concise and relatively short which is not a problem. However, the inadequacy can still be improved. At the moment there is a lack of geographical focus on the numerous points that you are mentioning. You mention less developed or developing or poor countries, but stop short of giving specific countries where this problem is happening. Are the water and sanitation problems happening in Somalia? or South Africa? or India? or Guatemala? I am therefore expecting you to contextualise the magnitude of these problems in specific countries. While you are allowed to specify certain generalisations as you have done, it would be more impactful if you stipulate a few case studies in specific countries to make us understand the global significance of this problem.  

(3) Somewhere in your introduction, can you please provide examples of LMIC countries? 

(4) In line 127, you have written as follows:  [According to many studies, women are primarily responsible for household hygiene  practices and water collection]: While the sentence is written well, you are expected to provide at least 4-5 sources to show at least that there were many studies indeed. 

(5) In line 28 you wrote as follows: [Furthermore, women took a more key role than men in household activities such as fetching water and caring for children]. Can you please enrich the context of your results by specifying specific countries where this finding was recorded. In that way you will be giving some context where this happened. Please make sure that in other set of results you provide such geographical context but don't do this excessively.  

(6) In line 251 you wrote as follows: [About 90% of slum toilets/baths/tube wells were shared by their community members ]. My question is 90% where? Where did this take place?

(7) While the results have been discussed adequately,  I find  your Conclusions section too short. Please add recommendations for future research.

(8) It is standard scientific practice in many scientific manuscripts that the caption of Tables is written above the table while it is written below Figures. In your manuscript, this problem applies to Figure 1, Figure 2, Figure 3a & 3b as well as Figure 4. Can you please correct accordingly. 

I hope that the comments are understandable and add to the improvement  of the manuscript. 

Author Response

Reviewer 2

RC: The manuscript makes an important literature contribution regarding the problems of water and sanitation in low-income or poor countries. This is because inadequate water, sanitation, and hygiene (WASH) among urban poor women is a major urban policy concern in low- and middle-income countries (LMICs).  Moreover, a gendered-perspective is provided in the manuscript which adds to the theoretical lens in which the research problem is examined. While the research reported is original and written relatively clear there are a few conceptual problems in a few areas.  A list of comments and suggestions is given for improvements and they are summarised as follows: 

AR: Thank You. We have revised the manuscript as suggested.

Comments and Suggestions for Improvement:

RC: (1) The original title is stated as 'A systematic review and gap analysis of opportunities and challenges in Water, Sanitation, and Hygiene (WASH) for urban poor in low- and middle-income countries during the COVID-19 pandemic through a gender lens'. Given the word count (i.e about 33 words) in this article, this title is too long. I suggest that you cut out the abbreviation (WASH) to at least try to shorten it and remove other superflous words.  

AR: We have revised the title as suggested.

RC: (2) Your introduction reads concise and relatively short which is not a problem. However, the inadequacy can still be improved. At the moment there is a lack of geographical focus on the numerous points that you are mentioning. You mention less developed or developing or poor countries, but stop short of giving specific countries where this problem is happening. Are the water and sanitation problems happening in Somalia? or South Africa? or India? or Guatemala? I am therefore expecting you to contextualise the magnitude of these problems in specific countries. While you are allowed to specify certain generalisations as you have done, it would be more impactful if you stipulate a few case studies in specific countries to make us understand the global significance of this problem.  

AR: We have extended the Introduction, in 48, lines 51-57.

RC: (3) Somewhere in your introduction, can you please provide examples of LMIC countries? 

AR: We have provided the LMICs countries.

RC: (4) In line 127, you have written as follows: [According to many studies, women are primarily responsible for household hygiene practices and water collection]: While the sentence is written well, you are expected to provide at least 4-5 sources to show at least that there were many studies indeed. 

AR: We have provided the references as suggested.

RC: (5) In line 128 you wrote as follows: [Furthermore, women took a more key role than men in household activities such as fetching water and caring for children]. Can you please enrich the context of your results by specifying specific countries where this finding was recorded. In that way you will be giving some context where this happened. Please make sure that in other set of results you provide such geographical context but don't do this excessively.  

AR: We have revised accordingly in lines 133-136.

RC: (6) In line 251 you wrote as follows: [About 90% of slum toilets/baths/tube wells were shared by their community members]. My question is 90% where? Where did this take place?

AR: We have provided the settings in lines 256-57.

RC: (7) While the results have been discussed adequately, I find your Conclusions section too short. Please add recommendations for future research.

AR: We added the recommendations for future research in conclusions lines 425-447.

RC: (8) It is standard scientific practice in many scientific manuscripts that the caption of Tables is written above the table while it is written below Figures. In your manuscript, this problem applies to Figure 1, Figure 2, Figure 3a & 3b as well as Figure 4. Can you please correct accordingly. 

AR: We have revised as suggested.

RC: I hope that the comments are understandable and add to the improvement of the manuscript.

AR: Thank You for your suggestions.